# Association of Physical Fitness, Screen Time, and Sleep Hygiene According to the Waist-to-Height Ratio in Children and Adolescents from the Extreme South of Chile

**DOI:** 10.3390/healthcare10040627

**Published:** 2022-03-27

**Authors:** Javier Albornoz-Guerrero, Fernanda Carrasco-Marín, Rafael Zapata-Lamana, Igor Cigarroa, Daniel Reyes-Molina, Olga Barceló, Guillermo García-Pérez-de-Sevilla, Sonia García-Merino

**Affiliations:** 1Departamento de Educación y Humanidades, Universidad de Magallanes, Punta Arenas 62000000, Chile; javier.albornoz@umag.cl; 2Centro de Vida Saludable, Universidad de Concepción, Concepción 4030000, Chile; fercarrasco@udec.cl; 3Escuela de Educación, Universidad de Concepción, Los Ángeles 4440000, Chile; rafaelzapata@udec.cl; 4Escuela de Kinesiología, Facultad de Salud, Universidad Santo Tomás, Los Ángeles 4440000, Chile; icigarroa@santotomas.cl; 5Facultad de Ciencias Sociales, Universidad de Concepción, Concepción 4030000, Chile; danielreyes@udec.cl; 6Department of Sports Sciences, Faculty of Sports Sciences, Universidad Europea de Madrid, 28670 Madrid, Spain; olga.barcelo@universidadeuropea.es; 7Department of Physiotherapy, Faculty of Sports Sciences, Universidad Europea de Madrid, 28670 Madrid, Spain; guillermo.garcia@universidadeuropea.es; 8Facultad de Ciencias de la Salud, Universidad Francisco de Vitoria, 28223 Madrid, Spain

**Keywords:** physical fitness, children, screen time, sleeping habits, body fat, adiposity

## Abstract

Objective: To analyze the perception of physical fitness, screen time, and self-reported sleep hygiene in children and adolescents (CA) from the extreme south of Chile and its associations with waist-to-height ratio (WtHr). Material and methods: An observational cross-sectional study was conducted in a sample of 594 schoolchildren from 5th to 8th grade of primary education, belonging to municipal educational establishments in the Magallanes region, Chile. Cardiorespiratory fitness was assessed through the 20-m shuttle run test, muscle strength through handgrip and the standing broad jump test, physical fitness perception through the International Fitness Scale, and central obesity through the waist-to-height index. In addition, sleep hygiene and screen time were measured. Results: More than 92% of CA spent more than two hours a day watching or using screens. In addition, CA with excess central adiposity had a lower perception of physical fitness, and lower muscle strength and cardiorespiratory fitness compared to CA with normal values of adiposity. Conclusions: CA of the present study spent a high number of hours watching or using screens and had poor sleep quality. In addition, excessive central adiposity was associated with lower physical fitness.

## 1. Introduction

Reducing obesity in children and adolescents (CA) is a major challenge for public health worldwide. It is estimated that more than 330 million children aged 5–19 years suffer from excess malnutrition [1]. In Chile, 25.4% of CA are obese/overweight. Specifically, the Magallanes region, where this study was carried out, belongs to an area in the extreme south of Chile characterized by extremely cold weather, which has 53.8% of overweight/obese CA in Chile [2]. Many authors have reported that presenting this condition at early ages has important short-, medium-, and long-term health consequences, such as high blood pressure, type 2 diabetes, dyslipidemia, sleep apnea, asthma, and atherosclerosis [3,4].

Within this context, the most commonly used indicator for the diagnostic of overweight and obesity in CA is the body mass index (BMI). However, BMI has been questioned due to its limitations in detecting adiposity in CA [5]. In contrast, waist-to-height ratio (WtHr) has shown to be a highly applicable [6], accurate anthropometric index to measure central adiposity and a strong predictor of arterial hypertension and type 2 diabetes [7,8,9].

Although several studies associating lifestyle with obesity use BMI, other authors have incorporated the WtHr as a marker of central adiposity [9], showing greater sensitivity to the association with physical fitness, muscle strength, and cognitive functioning [10]. Likewise, low physical activity levels, high amounts of screen time, and low sleeping hours have been associated with obesity assessed by the WtHr and above all during the SARS-CoV-2 lockdown [11,12,13].

Overweight/obesity is a multifactorial condition where lifestyle plays a key role in its development in CA [14]. In this regard, sleeping fewer hours than recommended has been associated with higher adiposity while adequate physical fitness seems to help reduce adiposity and has also been reported as a mediator of cognitive performance in CA [10,11,15,16].

Few studies have investigated the influence of extremely cold weather in controlling overweight/obesity in CA [17]. In this context, it has been reported that being overweight is more prevalent in territories with extremely low temperatures [18]. This type of climate favors physical inactivity and modifies hormonal production, such as increased ghrelin and cortisol, which increase appetite and lipid storage mechanisms [19]. Additionally, in countries with cold climates, daylight varies markedly along the seasons, with only 35–60 days a year of sunlight and the rest of the year with darkness for much of the day, with a negative impact on mental health and sleeping hours, reducing the hours of sleep in CA to seven hours per day or less [17,18,19,20,21].

To date, few studies have analyzed physical fitness and nutritional status in CA from territories with extremely cold climates. Furthermore, few have used WtHr as a marker of nutritional health and analyzed its association with lifestyle and physical fitness in CA. No research has been found studying the association between WtHr and physical fitness, screen time, and sleep habits in CA from territories with extremely cold climates. The present study is relevant because the Magallanes region has the highest prevalence of childhood overweight and obesity in Chile [22].

Additionally, it is a region that, due to geographic, climatic, and light-dark cycle conditions, does not favor systematic programs for the control of overweight, obesity, and the practice of physical activity. This study could make it possible to generate a profile of lifestyles (screen time and sleep hygiene) and perception of physical fitness of CA in a region with an extremely cold climate, being the first study of this kind conducted in South America.

This study aimed to analyze the perception of physical fitness, screen time, and self-reported sleep hygiene in CA from the extreme south of Chile and its associations with the WtHr.

## 2. Material and Methods

### 2.1. Study Design

An observational, cross-sectional study was conducted using data from the health survey of CA from the extreme south of Chile (2019). The variables analyzed were screen time, physical fitness, sleep hygiene, and WtHr.

### 2.2. Participants

CA from the second cycle of primary education (5th to 8th year) from three public educational establishments in Punta Arenas city (region of Magallanes and Chilean Antarctica) were invited to participate. A representative sample of 615 CA completed all the evaluations after acquiring the signed consent of the parents/legal guardians. A total of 14 CA were excluded for not having the signed informed consent, 4 CA did not participate due to inability to perform the physical tests, and 3 CA did not complete all the evaluations, so the final sample was 594 CA. The sample calculation was obtained considering 50% of heterogeneity, a margin of error of 5%, and a confidence level of 95%.

### 2.3. Implementation

The municipal corporation of Punta Arenas (CORMUPA) and the research team signed a collaboration agreement. Then, three educational establishments were randomly selected. The management team, the teachers, and the research team planned the study design and its execution together.

The research team was in charge of hiring a kinesiologist, a nutritionist, a psychologist, and a physical education teacher. Then, they conducted a short training session on assessment instruments to reduce the risk of inter-rater bias. The data collection was carried out in the first semester of the year 2020 within the educational establishments in spaces adapted for each test and test. All the tests were carried out within the class schedule and on the same day. Families, directors, teachers, and students were informed about the purpose of the study and agreed to collaborate in it. All schoolchildren who participated in the study gave their consent and their parents and legal guardians signed their informed consent. The project was approved by the Ethics Committee of the south-central macro zone of the Universidad Santo Tomás de Chile, code number 96–20, and all procedures were carried out following the Declaration of Helsinki and Singapore.

### 2.4. Variables

#### 2.4.1. Grouping Variable

Waist-to-height ratio (WtHr): WtHr is a specific index highly correlated with visceral adiposity. Waist circumference was measured taking as reference the equidistant point between the last non-floating rib and the iliac crest. Height was measured in centimeters. Then, WtHr was calculated by dividing waist (in cm) by height (in cm). Pediatric reference values are ≤0.47 normal adiposity; 0.47–0.50 moderate adiposity; and >0.50 excess adiposity [23,24].

#### 2.4.2. Physical Fitness

Cardio-respiratory fitness: The 20-m shuttle run test was used, measuring speed (m/s), and number of paliers, following the protocol and reference values given by Alpha Fitness for schoolchildren [25].

Muscle strength: handgrip strength dominant hand (digital dinamometer Smedley Baseline) and the standing broad jump test were performed, using the reference values given by Alpha Fitness for schoolchildren [26].

Physical fitness perception (IFIS): Evaluation of self-perception of the physical condition. This survey has five questions focused on knowing the self-perception of general physical fitness, cardio-respiratory fitness, muscle strength, flexibility, speed, and agility [27,28].

Nutritional status: The bodyweight of the schoolchildren was measured with a SECA^®^ brand digital scale (model 804, USA), the waist circumference was measured while standing with a 1.5 m tape, and the height was measured with a SECA^®^ brand portable stadiometer (model 213, USA). These measures were performed according to the standardized procedures described by the International Society of Film anthropometry (ISAK) [29] and according to the Habicht method [30]. Subsequently, the BMI was calculated to obtain the BMI/age indicator, and to classify the nutritional status of each child according to sex, considering as malnutrition a standard deviation (SD) ≥ −2, risk of malnutrition SD ≥ −1, and normality between 0.99 and −0.99 SD. Overweight, obesity, and severe obesity were determined by values of ≥1 SD, ≥2 SD, and ≥3 SD, respectively [31].

#### 2.4.3. Lifestyle

Screen time: A self-reported survey was applied by adding the responses (hours) of three questions: “How many hours a day do you usually watch television?”; “How many hours a day do you usually play video games on a tablet, computer, or cell phone?”; and “How many hours a day do you usually use a tablet, computer, or cell phone for purposes other than gaming, for example, email, chats, social networks, internet, or doing homework?” [32,33].

Sleep hygiene: The questionnaire used to assess sleep habits was the Sleep Self-Report (SSR) in its Spanish version. Each item has a 3-point scale to indicate the frequency of each habit: usually (2 = 5 to 7 times a week), sometimes (1 = 2 to 4 times a week), and rarely (0 = never or once a week). The questionnaire consists of 19 questions (three of them provide additional information but are not included in any subscale), grouped into 7 for sleep quality, 6 for sleep-related anxiety, 4 for rejection of sleep, 3 for routines to go to sleep, and 16 for the score total [33,34].

In addition, sociodemographic values such as sex (male/female), age (9–12 years/13–15 years), school grade (fifth grade/six grade/seventh grade/eight grade), place of residence (urban/rural), and if they belong to the school integration program (yes/no) were measured.

### 2.5. Statistical Analysis

Data were analyzed with the statistical software SPSS 25.0 (Windows, SPSS Inc., Chicago, IL, USA). Continuous variables were presented as mean and standard deviation, and categorical as percentages. After performing the Kolmogorov–Smirnov test to assess normality and the Levene’s test to assess homogeneity, all the variables showed a normal distribution and homogeneity of variances, so parametric statistics were used. Then, to establish an association between categorical variables, the Chi-square test was used. To establish differences between the three groups (normal adiposity, moderate adiposity, and excess adiposity) a one-factor ANOVA test was performed (level of adiposity). To determine differences between groups, a post hoc analysis (Bonferroni) was used. The association of adiposity and physical fitness were investigated using linear regression analyses. Data were presented as coefficient and its 95% CI. Multivariable models were adjusted for relevant confounding variables, selected from descriptive Table 1: Model 0—unadjusted; and Model 1—adjusted socio-educational and lifestyle variables (age, sex, school grade, special needs integration program, nutrition status, bedtime routine). Significance was set at the level of *p* < 0.05.

## 3. Results

Table 1 presents the physical fitness of CA according to socio-educational characteristics and lifestyle. In this sample of 594 CA, 58.8% of the participants were male, 61.3% were aged between 9 and 12 years, 83.4% did not belong to the school integration program, and 97.8% lived in urban places. As might be expected, boys performed better on tests of muscle strength (*p* < 0.001) and aerobic capacity (*p* < 0.001) than girls. In addition, children aged 13–15 years had better performance on tests of muscle strength (*p* < 0.001) and aerobic capacity (*p* < 0.01) than children aged 9–12 years. Furthermore, CA that belong to school integration programs had a lower standing broad jump test (*p* < 0.05) performance, compared with other CA. Moreover, CA with healthy adiposity had better performance in the standing broad jump test (*p* < 0.001) and the 20 m shuttle run test (*p* < 0.001) than children with overweight/obesity. Finally, children without sleeping problems had higher handgrip strength (*p* < 0.05) than children with sleeping problems.

Table 2 shows screen time, sleep hygiene, and perception of physical fitness according to central adiposity. A high percentage (>92%) of CA spent more than 2 h a day watching or using screens. In addition, 11–18% of the CA presented bedtime routine problems, anxiety related to sleep, or refusal to sleep, and 18–24% of the CA had poor sleep quality. All of these sleeping problems were independent of central adiposity. In contrast, adiposity was associated with the perception of physical fitness, as children with excessive adiposity had a lower perception of their general physical fitness (*p* < 0.001), cardiorespiratory fitness (*p* < 0.001), speed/agility (*p* < 0.001), and flexibility (*p* < 0.001).

Table 3 shows the cardiorespiratory fitness and muscle strength according to adiposity. There were significant differences in the standing broad jump test (*p* < 0.001) and in the 20 m shuttle run test (speed and shuttle) (*p* < 0.001) between CA with normal adiposity and CCA with moderate or excess adiposity. CA with normal adiposity had a better physical performance in all the tests performed.

Table 4 shows the association between cardio-respiratory fitness and muscle strength with the adiposity category. In the non-adjusted model, the CA with normal adiposity presented a better performance in the standing broad jump test (*p* < 0.001) compared to those who had excessive adiposity. Along the same lines, CA with normal (*p* < 0.001) and moderate (*p* < 0.001) adiposity presented a better performance in the 20 m shuttle run test than children with excessive adiposity. The same association was found (Table 4) once the model was adjusted for socio-educational and lifestyle variables such as age, sex, grade, participation in a school integration program, BMI, and bedtime routine (Table 1).

## 4. Discussion

This novel research suggests that a high percentage of CA of the south of Chile spend more than 2 h a day using or watching screens and have a low sleep quality. Children with excessive adiposity have a poor perception of physical fitness and poor cardiorespiratory fitness and muscle strength compared with children with normal adiposity.

The participants of our study spent too much screen time and not enough hours sleeping, regardless of their adiposity. This differs from what was expected since the evidence shows that high amounts of screen time and insufficient sleep are associated with obesity [11]. A study carried out on children in India found that poor sleep quality and low physical activity levels were associated with an increased risk of obesity, while high exposure to screens and low physical activity were associated with a higher BMI and WtHr [11]. The present study took place in a cold geographical area, which could have a fundamental role in metabolic processes such as the sleep cycle, hunger, and body temperature [35]. The evidence suggests that external conditions such as cold weather could be associated with higher screen time, less physical activity, and less exposure to sunlight, which may explain the decrease in sleeping hours, affecting the sleep cycle in all children [36,37].

On the other hand, it was observed that children with excessive adiposity reported a poor perception of physical condition more frequently than children with normal fat mass. In this regard, the perception of physical fitness is part of the multidimensional construct of physical self-concept [38]. In this sense, the role that fat has in the physical self-concept of children is not entirely clear. However, the direct association between physical self-concept and physical activity practice in CA is well documented [39,40]. In this line, Fernández-Bustos et al. conducted a study with CA aged 12–17 years [41], where they proposed a mediation model based on well-known theoretical models of physical self-concept [42,43]. They reported a direct effect of the fat mass on the body image and a mediating effect of fat mass and body image on physical self-concept [41].

The mediation model described above provides a plausible explanation for the findings of this study regarding the relationship between excessive fat mass and poor perception of physical fitness in children from the extreme south of Chile. These results evidence how environments with extreme cold weather favor excessive adiposity in CA [17,44,45]. Therefore, children from the extreme south of Chile could be in an environment that makes it hard to practice physical activity and do not favor physical self-concept.

CA with normal adiposity showed a higher aerobic capacity and muscle strength compared to those with excessive adiposity. These results are in line with the extensive existing evidence, where an inverse relationship between BMI and physical fitness is described [46,47,48]. Specifically, studies carried out in Chile indicate that overweight/obese children are the ones who present the worst aerobic capacity and muscle strength capacity [49,50,51]. However, BMI and physical performance do not show a linear relationship. Children with low weight also have lower physical fitness than those with adequate weight. In this sense, in a study carried out in 73,561 adolescents aged 13–15 years from Chile and Colombia, those with low or high BMI presented worse aerobic capacity than their peers with an adequate BMI [52].

In this context, we have previously reported on differences in anthropometric indicators according to the physical fitness of CA who live in the extreme south of Chile [23]. Overweight/obese students with higher muscle strength had a healthier waist circumference and WtHr [23].

### 4.1. Strengths and Limitations

This study has several strengths: a representative sample of CA belonging to educational centers in a city in the extreme south of Chile; the WtHr used as a grouping variable of the fat mass, which provides originality, relevance, and pertinence; and the context of childhood obesity within a territorial context of extreme cold weather. This study provides useful information about lifestyles and physical fitness of the CA for teachers and health professionals, particularly those who work in extremely cold territories.

The present study is a pioneer in Chile and South America in characterizing lifestyles, physical fitness, and body composition in children from territories with extremely cold climates. In addition, according to the literature studied, it is the first study that analyzes WtHr and its association with physical fitness, screen time, and sleep hygiene in children from territories with extremely cold climates.

However, this study has some limitations. For example, the variables of screen time and perception of physical fitness used self-reporting instruments so that the responses of the CA were subject to their mood, understanding, and even the maturity to understand the relevance of their responses. Nevertheless, the teachers who used the surveys answered all the doubts that the children had. This method of collecting information is widely used in the literature [32] and allows collecting data from a large sample in a short time. In addition, the type of study design only establishes relationships between adiposity and physical fitness and lifestyles but does not imply causality. In addition, although a multivariate analysis was used to evaluate the influence of covariates on the association between physical fitness and adiposity, there may be other variables that were not measured and that could interfere (example: type of diet, physical activity levels).

### 4.2. Practical Implications and Future Lines of Investigation

To date, there is little evidence about childhood overweight/obesity in territories with extremely cold climates. In the present study, children who had excessive adiposity had a low physical fitness perception and poor physical fitness [18]. In addition, climatic conditions such as extremely cold weather modify the lifestyles of CA, forcing them to spend more hours in closed spaces [44], leading to increased screen time, sleeping problems, anxiety, appetite, and fat percentage [11]. On the other hand, sleep and overweight/obesity are highly linked, especially in adverse weather conditions in countries with characteristics such as longer days in summer, many hours of darkness in winter, and low temperatures in winter [14].

In addition, this study shows the relevance of lifestyle and environmental factors such as sleep, screen time, physical activity, and extremely cold weather concerning overweight and obesity in children. These results can be used in educational centers and health centers to analyze how healthy lifestyles are being promoted and how weight control is being addressed in CA.

Future studies should further investigate the consequences of childhood obesity on the physical, mental, and emotional health of children living in extremely cold regions. Moreover, future research should focus on the risk factors leading to childhood obesity and use the WtHr as an adiposity marker.

As future lines of research, many studies use BMI, waist circumference, and WtHr as predictors of cardiovascular risk and physical fitness in children [29,30,31]. However, BMI has been questioned due to the limitations to detect adiposity in CA [5], so WtHr should be used instead. WtHr has been shown to be accurate in measuring abdominal fat predicting chronic diseases such as type 2 diabetes mellitus [7].

## 5. Conclusions

In the extreme south of Chile, children and adolescents spend a high number of hours watching or using screens and indicate having a low sleep quality. In addition, children with excessive central adiposity have a poor perception of their physical fitness and concordantly have worse cardio-respiratory fitness and muscle strength compared to those with normal central obesity. These results are independent of socio-educational and lifestyle variables such as age, sex, school grade, participation in a school integration program, BMI, and bedtime routine.

## Figures and Tables

**Table 1 healthcare-10-00627-t001:** Physical fitness according to socio-educational characteristics and lifestyle.

		Fitness			
		Muscle Strength		Cardiorespiratory Fitness	
		Handgrip Strength Dominant Hand (kg)	Standing Broad Jump Test (cm)	20 m Shuttle Run Test (Speed)	20 m Shuttle Run Test (Palier)
Variables	N (%)	Mean [95% CI]	Mean [95% CI]	Mean [95% CI]	Mean [95% CI]
**Socio-educational variables**					
Sex					
Male	302 (50.8%)	24.6 [23.7; 25.4] ***	128.5 [125.2; 131.9] ***	2.57 [2.56; 2.59] ***	31.2 [29.3; 33.0] ***
Female	292 (49.2%)	21.6 [21.0; 22.1]	110.8 [108.0; 113.6]	2.52 [2.50; 2.53]	24.7 [23.3; 26.2]
Age (years)					
9–12 years	364 (61.3%)	21.2 [20.6; 21.7] ***	114.4 [111.9; 116.9] ***	2.54 [2.52; 2.55]	26.9 [25.5; 28.4] **
13–15 years	230 (38.7%)	26.1 [25.2; 27.0]	128.4 [124.2; 132.5]	2.56 [2.54; 2.58]	29.7 [27.6; 31.7]
Grade					
Fifth grade	166 (27.9%)	19.3 [18.6; 20.0] ***	111.0 [107.8; 114.1] ***	2.53 [2.51; 2.55]	26.1 [24.0; 28.1]
Sixth grade	153 (25.8%)	22.1 [21.3; 22.8]	116.8 [112.8; 120.8]	2.55 [2.53; 2.57]	28.1 [25.8; 30.5]
Seventh grade	123 (20.7%)	24.1 [23.1; 25.1]	119.5 [114.0; 125.0]	2.53 [2.51; 2.56]	27.2 [24.6; 29.8]
Eightth grade	152 (25.6%)	27.3 [26.2; 28.5]	132.7 [127.5; 137.9]	2.57 [2.54; 2.59]	30.5 [27.9; 33.2]
School Integration Program					
Yes	99 (16.8%)	23.7 [22.4; 25.0]	125.1 [119.5; 130.7] *	2.57 [2.54; 2.59]	30.0 [27.0; 33.0]
No	495 (83.4%)	23.0 [22.4; 23.5]	118.7 [116.2; 121.2]	2.54 [2.53; 2.55]	27.6 [26.3; 28.9]
Place of residence					
Urban	581 (97.8%)	23.1 [22.6; 23.6]	120.0 [117.7; 122.3]	2.55 [2.53; 2.56]	28.0 [26.8; 29.2]
Rural	13 (2.2%)	21.9 [18.4; 25.5]	112.3 [97.8; 126.8]	2.53 [2.46; 2.60]	26.2 [18.7; 33.8]
**Lifestyle variables**					
Nutritional status					
Normal weight	162 (27.3%)	22.5 [21.5; 23.4]	130.4 [126.0; 134.7] ***	2.59 [2.57; 2.61] ***	33.8 [31.4; 36.2] ***
Overweight	421 (72.7%)	23.4 [22.8; 24.1]	115.6 [113.0; 118.2]	2.52 [2.51; 2.54]	25.5 [24.2; 26.8]
Screen time					
<2 h/day	33 (7.7%)	22.4 [20.3; 24.5]	117.9 [107.2; 128.5]	2.52 [2.48; 2.57]	24.5 [20.2; 28.7]
≥2 h/day	548 (92.3%)	23.1 [22.5; 23.6]	120.1 [117.8; 122.5]	2.55 [2.53; 2.56]	28.2 [26.9; 29.4]
Bedtime routine					
Adequate	514 (86.5%)	23.3 [22.7; 23.9] *	120.0 [117.5; 122.5]	2.55 [2.53; 2.56]	28.2 [26.9; 29.5]
With problems	80 (13.5%)	21.7 [20.5; 22.9]	118.4 [112.4; 124.5]	2.54 [2.51; 2.57]	26.7 [23.6; 29.8]
Sleep anxiety					
Without anxiety	512 (86.2%)	23.2 [22.7; 23.7]	119.9 [117.4; 122.4]	2.54 [2.53; 2.55]	27.7 [26.4; 29.0]
With anxiety	82 (13.8%)	22.4 [20.9; 23.8]	119.1 [113.1; 125.1]	2.57 [2.54; 2.59]	29.5 [26.4; 32.6]
Sleep quality					
Adequate	456 (76.8%)	23.3 [22.7; 23.8]	120.4 [117.8; 123.1]	2.54 [2.53; 2.55]	27.5 [26.2; 38.9]
Poor	138 (23.2%)	22.5 [21.4; 23.6]	117.8 [113.3; 122.4]	2.56 [2.54; 2.58]	29.5 [26.9; 32.0]
Refusal to sleep					
No	503 (84.8%)	23.3 [22.7; 23.8]	120.0 [117.5; 122.5]	2.54 [2.53; 2.56]	27.9 [26.6; 29.1]
Yes	91 (15.2%)	22.0 [20.8; 23.3]	118.9 [113.2; 124.6]	2.55 [2.52; 2.58]	28.7 [25.5; 31.9]

The qualitative variables are presented in absolute and percentage frequency and the quantitative ones are presented in mean and their respective CI 95%. *** = the differences are significant at *p* < 0.001, ** = the differences are significant at *p* < 0.01. * = the differences are significant at *p* < 0.05. n = 594.

**Table 2 healthcare-10-00627-t002:** Screen time, bedtime routine, and physical fitness perception.

Variables	Normal Adiposity N (%)	Moderate Adiposity N (%)	Excess Adiposity N (%)	*p*-Value
**Screen Time and Sleep Hygiene**				
Screen time				
<2 h/day	20 (7.5%)	6 (6.5%)	7 (3.1%)	0.106
≥2 h/day	246 (92.5%)	86 (93.5%)	216 (96.9%)	
Bedtime routine				
Adequate	239 (87.9%)	80 (84.2%)	195 (85.9%)	0.627
With problems	33 (12.1%)	15 (15.8%)	32 (14.1%)	
Sleep anxiety				
Without anxiety	240 (88.2%)	78 (82.1%)	194 (85.5%)	0.303
With anxiety	32 (11.8%)	17 (17.9%)	33 (14.5%)	
Sleep quality				
Adequate	205 (75.4%)	77 (81.1%)	174 (76.7%)	0.528
Poor	67 (24.6%)	18 (18.9%)	53 (23.3%)	
Refusal to sleep				
No	233 (85.7%)	82 (86.3%)	188 (82.8%)	0.605
Yes	39 (14.3%)	13 (13.7%)	39 (17.2%)	
Total sleep score (SSR)				
Healthy sleep	232 (85.3%)	78 (82.1%)	186 (81.9%)	0.557
With sleep problems	40 (14.7%)	17 (17.9%)	41 (18.1%)	
**Physical Fitness Perception (IFIS)**				
General physical fitness				
Very good	72 (26.5%)	10 (10.5%)	30 (13.2%)	<0.001
Good	81 (29.8%)	28 (29.5%)	48 (21.1%)	
Fair	90 (33.1%)	36 (37.9%)	81 (35.7%)	
Poor	23 (8.5%)	18 (18.9%)	55 (24.2%)	
Very poor	6 (2.2%)	3 (3.2%)	13 (5.7%)	
Cardiorespiratory fitness				
Very good	72 (26.5%)	9 (9.5%)	25 (11.0%)	<0.001
Good	80 (29.4%)	22 (23.2%)	41 (18.1%)	
Fair	83 (30.5%)	30 (31.6%)	68 (30.0%)	
Poor	34 (12.5%)	30 (31.6%)	72 (31.7%)	
Very poor	3 (1.1%)	4 (4.2%)	21 (9.3%)	
Muscle strength				
Very good	58 (21.3%)	12 (12.6%)	31 (13.7%)	0.124
Good	79 (29.0%)	25 826.3%)	62 (27.3%)	
Fair	79 (29.0%)	38 (40.0%)	70 (30.8%)	
Poor	46 (16.9%)	18 (18.9%)	52 (22.9%)	
Very poor	10 (3.7%)	2 (2.1%)	12 (5.3%)	
Velocity/agility				
Very good	55 (20.2%)	11 (11.6%)	27 (11.9%)	<0.001
Good	88 (32.4%)	31 (32.6%)	43 (18.9%)	
Fair	86 (31.6%)	34 (35.8%)	87 (38.3%)	
Poor	34 (12.5%)	13 (13.7%)	58 (25.6%)	
Very poor	9 (3.3%)	6 (6.3%)	12 (5.3%)	
Flexibility				
Very good	55 (20.2%)	9 (9.5%)	19 (8.4%)	<0.001
Good	63 (23.2%)	33 (34.7%)	37 (16.3%)	
Fair	76 (27.9%)	22 (23.2%)	66 (29.1%)	
Poor	65 (23.9%)	22 (23.2%)	70 (30.8%)	
Very poor	13 (4.8%)	9 (9.5%)	35 (15.4%)	

Qualitative variables are presented in absolute and percentage frequency. To establish an association between categorical variables, the Chi-square test was used. Significance was set at the level of *p* < 0.05. n = 594.

**Table 3 healthcare-10-00627-t003:** Cardiorespiratory fitness and muscle strength according to adiposity.

Variables	Normal Adiposity	Moderate Adiposity	Excess Adiposity	One Way ANOVA	
	Mean [95% CI]	Mean [95% CI]	Mean [95% CI]	F	*p*-Value
**Muscle strength**					
Handgrip strength dominant hand (kg)	23.1 [22.4; 23.8]	22.65 [21.28; 24.02]	23.24 [22.37; 24.12]	0.292	0.747
Standing broad jump test (cm)	128.6 [125.2; 132.0] ^a^	116.9 [110.9; 122.9] ^b^	110.5 [107.3; 113.6] ^b^	28.410	<0.001
**Cardiorespiratory fitness**					
20 m shuttle run test (speed) (m/s)	2.60 [2.58; 2.61] ^a^	2.55 [2.53; 2.57] ^b^	2.48 [2.47; 2.50] ^c^	51.692	<0.001
20 m shuttle run test (n° palier)	33.86 [32.0; 35.72] ^a^	26.49 [24.01; 28.98] ^b^	21.57 [20.02; 23.12] ^c^	50.120	<0.001

Each group are presented in mean and their respective CI 95%. The statistical analysis was carried out through a one-way ANOVA. In the same row, ^abc^ different lowercase letters indicate statistically significant differences between groups. (One-way ANOVA and Bonferroni post hoc). Significance was set at the level of *p* < 0.05. n = 594.

**Table 4 healthcare-10-00627-t004:** Association between physical fitness and the adiposity category.

	Model 0		Model 1	
	Unadjusted Model		Adjusted Model	
Variables	β_i_ [95% CI]	*p*-Value	β_i_ [95% CI]	*p*-Value
**Muscle strength**				
Handgrip strength dominant hand				
Normal adiposity	−0.15 [−1.27; 0.98]	0.799	−0.11 [−1.26; 1.04]	0.849
Moderate adiposity	−0.59 [−2.12; 094]	0.446	−0.83 [−2.14; 0.48]	0.213
Exceed adiposity	Reference		Reference	
Standing broad jump test				
Normal adiposity	18.18 [13.39; 22.97]	<0.001	14.17 [8.98; 19.36]	<0.001
Moderate adiposity	6.42 [−0.10; 12.93]	0.054	5.75 [−0.16; 11.66]	0.057
Exceed adiposity	Reference		Reference	
**Cardiorespiratory fitness**				
20 m shuttle run test (speed)				
Normal adiposity	0.11 [0.09; 0.13]	<0.001	0.11 [0.08; 0.13]	<0.001
Moderate adiposity	0.06 [0.03; 0.09]	<0.001	0.07 [0.04; 0.10]	<0.001
Exceed adiposity	Reference		Reference	
20 m shuttle run test (n° palier)				
Normal adiposity	12.29 [9.86; 14.72]	<0.001	11.49 [8.69; 14.30]	<0.001
Moderate adiposity	4.93 [1.63; 8.22]	0.003	5.24 [2.04; 8.43]	0.001
Exceed adiposity	Reference		Reference	

The data are presented as β levels with their corresponding CI 95%, according to the physical fitness. The statistical analysis was conducted through a multiple linear regression analysis. Exceed fat was considered the reference values (ref.). The statistical analyses were progressively adjusted. Model 0: non adjusted, Model 1: adjusted for socio-educational and lifestyle variables (age, sex, school grade, special needs integration program, nutrition status, bedtime routine). Significance was set at the level of *p* < 0.05. n = 594.

## Data Availability

Data are available upon request due to ethical and privacy restrictions.

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
