# Peer review of "Association of Physical Fitness, Screen Time, and Sleep Hygiene According to the Waist-to-Height Ratio in Children and Adolescents from the Extreme South of Chile"

_healthcare, 2022, doi:10.3390/healthcare10040627_

Round 1
Reviewer 1 Report
The data are not essentially knew, but the study has been well done. It is important to know whether the questionnaire was answered by the children or by their parents. Eventually the authors could have done both. By whom was the screen time measured. Did the children know that the screen time would be measured or was this done in a blind fashion?
Author Response
Dear reviewer,
According to your decision of Minor Revisions, we appreciate the opportunity to revise and re-submit our manuscript Healthcare-1641159, entitled “Association of physical fitness, screen time and sleep hygiene according to the waist-to-height ratio in children and adolescents from the extreme south of Chile". We trust that you will find the current version informative to your readership and acceptable for publication.
Our manuscript has been changed according to the required recommendations. The changes made are highlighted in yellow.
Sincerely, the Authors.

Reviewer 2 Report
I would like to thank you for the opportunity to review this article and congratulate the authors for this work. For me, as a physical educator, this topic is very important and has a lot of value. I really enjoyed reading this manuscript. Below are my suggestions and at the end my recommendation.
This paper analyzed the relationship of waist-height Fagio with fitness level, screen time and sleep habits in children and adolescents in the upper end of Chile.
Title: The title is concrete, representative and indicative of the problem investigated in the manuscript.
Abstract: The abstract is clear and complies with the general rules for writing a good abstract. However, in the third line the acronym WtHr is mentioned to refer to Waist-to-Height Ratio. Since this is the first time it is mentioned in the manuscript, it should read: Waist-to-Height Ratio (WtHr).
It is also suggested to describe a little better the tests used to measure sleep hygiene and screen time. I consider this section to be the most important, as it will be read more times than the manuscript itself.
Introduction
As I have already mentioned, I consider this research to be extremely important in contributing to the field of Physical Education and health. I do not disagree with the authors' justifications and I read many very good and current arguments and the context in which the study was developed is detailed. In addition, the relevance of the study is argued by justifying the prevalence of childhood overweight and obesity in the region where the study was conducted. It is suggested to the authors that from the stated objective, the research questions that help to carry out the research and the discussion from the findings in which the study variables, the study population and the expected result appear are highlighted.
Material and method.
Research design: It is suggested to expand this section. Data from the 2019 Chilean far south health survey are mentioned, but more information is missing.
Participants. Correct. The characteristics of the subjects included are described (sample selection criteria (inclusion and exclusion).
Procedures: Correct.
Statistical analysis: To help a reader unfamiliar with statistical studies to better understand the manuscript, it is suggested to define the categorical and continuous variables analyzed.
Results:
It is suggested to include a sample characterization table 1 to help the reader know the characteristics of the participants.
The rest of the results are correctly shown and are easy to read and simple for a scholar not accustomed to quantitative methodology.
Discussion: Correct and supported by previous studies.
Conclusions: They are clear and provide an answer to the stated objectives. However, it is suggested, if the research question is finally introduced, to confirm the answer in this section.
I recommend that this manuscript undergo another round of revision after minor revisions.
Author Response

(The authors gave the same response as above.)

Reviewer 3 Report
First of all I would like to thank you for the opportunity to review your work. I think it is a very important and worrying issue among the world's population, especially among the youngest members of society.
I think that the work is correct in its entirety and that all the basic sections of the article are perfectly described.
However, I would recommend including recent studies on childhood overweight and obesity as well as on anthropometric measurements that have been carried out during the pandemic period. I believe that the problem has worsened during this period and could be a good complement to your introduction. The Introduction is the point that I think needs to be improved before final publication. Some interesting references in this regard are:
Ramos Álvarez, O.; Arufe Giráldez, V.; Cantarero Prieto, D.; Ibáñez García, A. Changes in Physical Fitness, Dietary Habits and Family Habits for Spanish Children during SARS-CoV-2 Lockdown. Int. J. Environ. Res. Public Health 2021, 18, 13293. https://doi.org/10.3390/ijerph182413293
Ramos-Álvarez, O.; Arufe-Giráldez, V.; Cantarero-Prieto, D.; Ibáñez-García, A. Impact of SARS-CoV-2 Lockdown on Anthropometric Parameters in Children 11/12 Years Old. Nutrients 2021, 13, 4174. https://doi.org/10.3390/nu13114174
Thank you very much!
Author Response

(The authors gave the same response as above.)
